# FRETpredict: a Python package for FRET efficiency predictions using rotamer libraries
Daniele Montepietra[1,2,5], Giulio Tesei [3,5], João M. Martins [3], Micha B. A. Kunze [3],
Robert B. Best [4] ✉ & Kresten Lindorff-Larsen [3] ✉

Förster resonance energy transfer (FRET) is a widely-used and versatile technique for the structural characterization of biomolecules. Here, we introduce FRETpredict, an easy-to-use Python software to predict FRET efficiencies from ensembles of protein conformations. FRETpredict uses a rotamer library approach to describe the FRET probes covalently bound to the protein. The software efficiently and flexibly operates on large conformational ensembles such as those generated by molecular dynamics simulations to facilitate the validation or refinement of molecular models and the interpretation of experimental data. We provide access to rotamer libraries for many commonly used dyes and linkers and describe a general methodology to generate new rotamer libraries for FRET probes. We demonstrate the performance and accuracy of the software for different types of systems: a rigid peptide (polyproline 11), an intrinsically disordered protein (ACTR), and three folded proteins (HiSiaP, SBD2, and MalE). FRETpredict is open source (GPLv3) and is available at github.com/KULL-Centre/FRETpredict and as a Python PyPI package at pypi.org/project/FRETpredict.

Förster resonance energy transfer (FRET) is a well-established technique to measure distances and dynamics between two fluorophores[1,2]. Single-molecule FRET (smFRET) and ensemble FRET have been broadly used to study protein and nucleic acid conformational states and dynamics[3–5], binding events[6,7], and intramolecular transitions[8,9]. The high spatial (nm) and temporal (ns) resolutions enable FRET experiments to uncover individual species in heterogeneous and dynamic biomolecular complexes, as well as transient intermediates[10–15].

In a typical smFRET experiment on proteins, two residues are labeled with a donor and an acceptor FRET probe, respectively. While the FRET probes may sometimes be fluorescent proteins, they are more commonly organic molecules optimized for spectral and photophysical properties. Each such probe consists of a fluorophore and a linker, which can vary in length and is covalently attached to the protein[16]. For FRET to occur, the donor and acceptor fluorophores must have respective emission and absorption spectra that partially overlap, and the efficiency of the energy transfer depends on the proximity and relative orientation of the fluorophores.

Computational advancements, combined with enhanced sampling methods and approaches to coarse-grain, have enabled molecular dynamics (MD) simulations of biomolecules to explore time scales up to the millisecond or beyond[17–19]. Concomitantly, the molecular-level insights into protein structural dynamics provided by MD simulations are routinely employed to aid the interpretation of a multitude of experimental approaches, including FRET measurements[14,20]. Irrespective of whether the underlying protein structure is static or dynamic, the conformational ensembles of the fluorescent probes must be taken into account to accurately predict FRET efficiencies from MD simulations[21].

To model the conformational space of dyes attached to a protein, several methods have been developed[22,23]. At the low end of the spectrum of computational cost, the available volume (AV) method uses a coarse-grained description of the probe for predicting the geometric volume encompassing the conformational ensemble of the probe[24–26], achieving good accuracy with smFRET experimental data[13,15,16]. At the high end, MD simulations can be performed with explicit FRET probes[20,22,27–29], achieving high accuracy[18]. Although this approach provides unique insight into the

[1]Department of Chemical, Life and Environmental Sustainability Sciences, University of Parma, Parma 43125, Italy. [2]Istituto Nanoscienze - CNR-NANO, Center S3, via G. Campi 213/A, 41125 Modena, Italy. [3]Structural Biology and NMR Laboratory & the Linderstrøm-Lang Centre for Protein Science, Department of Biology, University of Copenhagen, Copenhagen DK-2200, Denmark. [4]Laboratory of Chemical Physics, National Institute of Diabetes and Digestive and Kidney Diseases, National Institutes of Health, Bethesda, MD 20892-0520, USA. [5]These authors contributed equally: Daniele Montepietra, Giulio Tesei.
✉ e-mail: robert.best2@nih.gov; lindorff@bio.ku.dk

motion of and interactions between protein and FRET probes, it is limited in its capability to sample the conformational space, particularly since the dye distribution changes with conformational changes of the biomolecule[23]. Such studies also depend on that the force field used for the fluorescent dyes fully being accurate and compatible with the force field used for the biomolecule[22,28,30]. Furthermore, comparison with studies that integrate results from multiple pairs of probe positions requires running independent MD simulations for each probe pair. Somewhere in the middle of the scale of computational cost and resolution is the rotamer library approach (RLA), where multiple rotamer conformations of the FRET probe are placed at the labeled site of a protein conformation, and the statistical weight of each conformer is estimated using a simplified potential[31]. The advantage of the RLA over MD simulations with explicit FRET probes is that it reduces the computational effort, since the simulations required to generate a rotamer library for a new FRET probe only need to be performed once, and the resulting library can then be applied to many different systems. In addition, the simulated system is considerably smaller. Polyhach et al.[31] introduced the RLA in the context of electron paramagnetic resonance[32]. The RLA may, however, also be employed to predict FRET[22,33], in addition to double electron-electron resonance (DEER) and paramagnetic relaxation enhancement (PRE) nuclear magnetic resonance data[31,32,34,35].

In this work we introduce FRETpredict, an easy-to-use Python module based on the RLA that enables FRET efficiency calculations from protein conformational ensembles. We describe a general methodology to generate rotamer libraries for FRET probes and provide access to rotamer libraries for many commonly used dyes and linkers. We present case studies for proteins displaying different dynamics ranging from disordered proteins to flexible and relatively static folded proteins (ACTR, Polyproline 11, HiSiaP, SBD2, and MalE). We selected systems for which FRET data has been carefully measured and validated using independent methods. The systems cover sizes up to 370 residues (for MalE), showing that both FRETpredict and FRET experiments are applicable to large systems and distances.

## Results

### Rotamer libraries

Each FRET probe consists of two parts: the fluorescent dye, responsible for the FRET, and the linker, which comprises (i) a spacer, to distance the dye from the protein and (ii) a moiety to attach the probe covalently to the protein. For example, many of the most widely used probes can be purchased with maleimide (to link to Cys), N-hydroxysuccinimide (to link to Lys), or azide (for click chemistry) functional groups.

As detailed in Supplementary Note 1, we generated rotamer libraries through a series of clustering steps and, to further decrease their sizes, we filtered out low-populated cluster centers based on three different cutoffs. Briefly, the steps used to generate the rotamer libraries are (i) generation of the conformational ensemble of the FRET probe using MD simulations; (ii) selection of combinations of the most populated dihedral angles to generate the C1 set of cluster centers; (iii) assignment of trajectory frames to the C1 set based on the least-square deviations of the dihedral angles; (iv) average over the angles of the trajectories frames of each C1 cluster center to generate the C2 set of cluster centers; (v) assignment of trajectory frames to the C2 set based on the least-square deviations of the dihedral angles; (vi) filtering of clusters with populations lower than 10, 20, and 30 structures to generate the rotamer libraries referred to as large, medium, and small hereafter. In this work, we created rotamer libraries for AlexaFluor, ATTO, and Lumiprobe dyes with different linkers (Supplementary Figs. 1, 2 and 3), using the force fields developed by Graen et al.[36]. This selection of rotamer libraries of widely used FRET probes are made available as a part of the FRETpredict package.

To illustrate the extent to which the conformational ensemble of the probes is reduced upon the generation of the rotamer libraries, we plotted the projection on the $xy$-plane of the distance vectors between the C$\alpha$ atom and the central atom of the fluorophore (Fig. 1 and Supplementary Figs. 4, 5 and 6) of all the generated rotamer libraries. Compared to the unfiltered rotamer libraries (Supplementary Fig. 4), the distribution of fluorophore positions for the large rotamer libraries (cutoff = 10) are less isotropic and homogeneous, as evidenced by the deviation of the scatter plot from a circular shape. Unsurprisingly, the anisotropicity is increasingly more pronounced for the medium and small rotamer libraries which were obtained by filtering out clusters of less than 20 and 30 conformers, respectively (Supplementary Figs. 5 and 6).

The rotamer libraries of some FRET probes show pronounced anisotropy, illustrated by the deviation of the scatter plots from a circular shape (A48 L1R, A53 L1R, A56 L1R, A59 L1R, and A48 B1R). The observed anisotropy can be related to the length of the linker, and hence to its rotational degrees of freedom. For example, the rotamer library A48 C1R is more isotropic than A48 L1R because L1R is a shorter linker than C1R (Supplementary Fig. 1). On the other hand, a comparison between A48 L1R and T42 L1R suggests that the more isotropic nature of T42 might be due to the structure of the T42 fluorophore which effectively provides an extension to the linker length (Supplementary Fig. 2).

The RLA relies on a trade-off between thorough conformational sampling and computational cost, as the latter increases with the increased size of the library (Supplementary Fig. 7), which ideally should not exceed

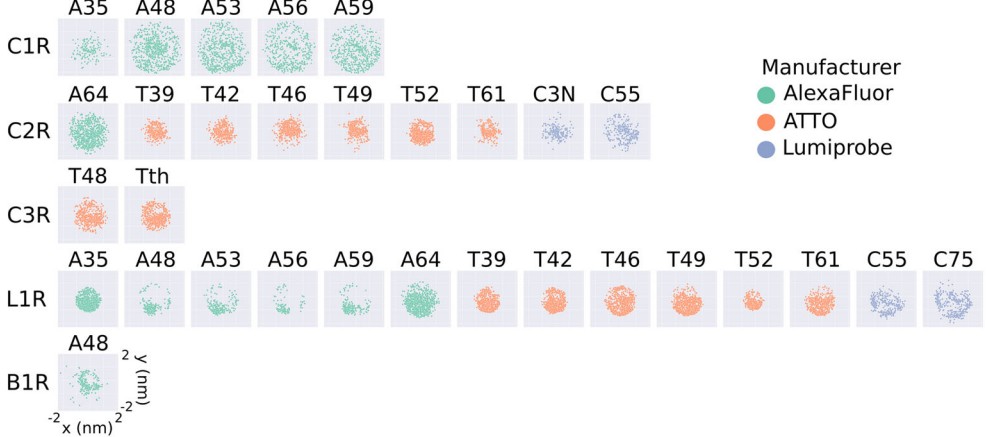

**Fig. 1 | 2D projections of the position of the fluorophore with respect to the C$\alpha$ atom.** Projection on the $xy$-plane of the distance vectors between the C$\alpha$ atom and the central atom of the fluorophore for the large rotamer libraries generated in this work, which typically contain hundreds of structures (Supplementary Fig. 7). The projections are obtained as the $x$ and $y$ coordinates of the central atom of the fluorophore (*O91* for AlexaFluor, *C7* for ATTO, and *C10* for Lumiprobe), after placing the C$\alpha$ atom at the origin. Each plot represents a different FRET probe, divided into rows according to linker type (C1R, C2R, C3R, L1R, B1R, from top to bottom), and colored according to the manufacturer (green for AlexaFluor, orange for ATTO, and blue for Lumiprobe).

~1000 rotamers. To provide an idea of the time differences involved in using rotamer libraries with different numbers of rotamers, we report the times required to calculate the FRET efficiencies for Polyproline 11 (Supplementary Table 1).

## FRETpredict algorithm

For each protein structure to be analysed—either individually or as an ensemble—the FRETpredict algorithm places the FRET probes at the selected protein sites independently of each other (Supplementary Note 2). Relative orientations and distances between the dyes are then computed for all combinations of the donor and acceptor rotamers. Further, nonbonded interaction energies between each rotamer and the surrounding protein atoms are calculated within a radius of 1.0 nm. Using these energies, statistical weights are first estimated for donor and acceptor rotamers independently and subsequently combined to calculate average FRET efficiencies (Fig. 2). The calculation of the average FRET efficiency are implemented assuming three different averaging regimes (detailed in Methods).

FRETpredict is written in Python and is available as a Python package. The `FRETpredict` class carries out the FRET efficiency predictions. The class is initialized with (i) a protein structure or trajectory (provided as `MDAnalysis Universe` objects[37]), (ii) the residue indices to which the fluorescent probes are attached, and (iii) the rotamer libraries for the fluorophores and linkers to be used in the calculation. The *lib/libraries.yml* file lists all the available Rotamer Libraries, along with necessary fluorophore information, including atom indices for calculating transition dipole moments and distances between fluorophores. As shown in the *Results* section, the calculations are triggered by the *run* function.

The main requirements are `Python 3.6-3.8` and `MDAnalysis 2.0`[37]. FRETpredict can be installed through the package manager `PIP` by executing

```
pip install FRETpredict
```

Tests predicting FRET data for the multidomain protein Hsp90[16] can be run locally using the test running tool `pytest`.

In the following, we showcase how FRETpredict can be used to calculate FRET efficiencies using different labels, different averaging schemes and different types and sources of protein/peptide conformations. Our goal here is not to discuss the biophysics of the individual systems, but rather to highlight the capabilities of FRETpredict.

## Case study 1: simulation trajectory of pp11

Polyproline 11 (pp11) has been described as behaving like a rigid rod, and was used as a "spectroscopic ruler" in the seminal paper by Stryer and Haugland[38]; subsequent work showed additional complexity[29,39–41]. The pp11 system is thus a classical example of the importance of comparing molecular models with FRET data. Here, we compared FRET efficiency values estimated using FRETpredict with reference values from experiments[39] and from extensive all-atom MD simulations of pp11 with explicit FRET probes[28]. For analyses with the RLA we removed these FRET probes to ensure that the conformational ensembles were comparable, and thus compared the different ways of representing the dyes (explicitly or via RLA). In both experiments and simulations, the terminal residues were labeled with AlexaFluor 488 - C1R (donor) and AlexaFluor 594 - C1R (acceptor), and the $R_0$ value was fixed to 5.4 nm. We used large rotamer libraries to estimate the FRET efficiency of pp11 in the three averaging regimes (Fig. 3 and Supplementary Table 2). We observe that the Dynamic regime best approximates the experimental value. As a convenient approach to calculate FRET efficiencies when there is no information about which averaging regime to use, we also calculate the average, $\langle E \rangle$, over the estimates of the static, dynamic, and dynamic+ regimes. Comparison with the reference values (Fig. 3 and Supplementary Table 2) indicates that FRETpredict yields predictions that are in slightly better agreement with experiments than MD simulations with explicit representation of the probes. This result suggests that the RLA provides relatively accurate FRET predictions and that MD simulations may not necessarily yield the most accurate result unless they are able to adequately sample the full range of dye conformations[42,43].

FRET efficiencies were calculated from the pp11 trajectory through the following lines of code:

```
from FRETpredict import FRETpredict
u = MDAnalysis.Universe("pp11.pdb", "pp11.xtc")
FRET = FRETpredict (protein=u, residues=[0, 12],
electrostatic=True,
        donor="AlexaFluor 488", acceptor="Alexa
        Fluor 594",
        libname_1="AlexaFluor 488 C1R cutoff10",
        libname_2="AlexaFluor 594 C1R cutoff10")
FRET.run()
FRET.reweight()
```

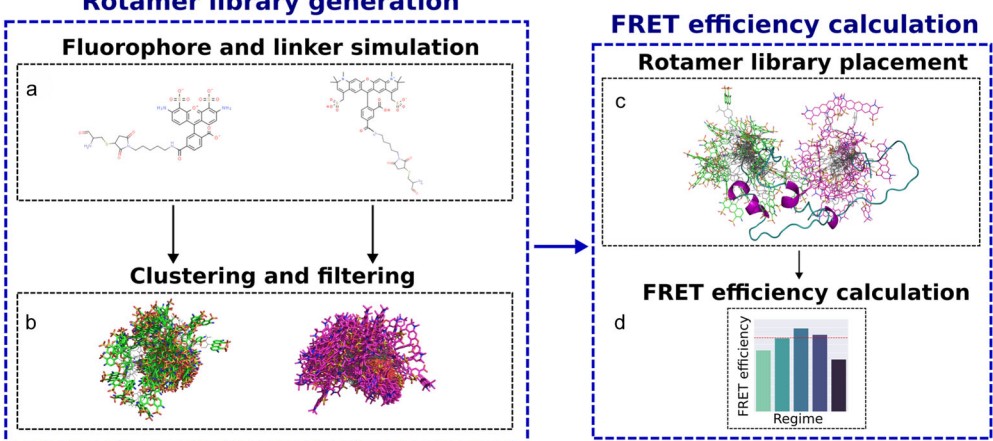

**Fig. 2 | Visual summary of the functionalities of FRETpredict.** FRETpredict consists of two main routines: rotamer library generation (**a, b**) and FRET efficiency calculation (**c, d**). **a** All-atom MD simulations of a free FRET probe in solution are performed to thoroughly sample the conformational ensemble of the probe. **b** The obtained conformations are clustered and the clusters are filtered by population size to generate the rotamer library of the FRET probe. **c** The rotamer libraries of the donor and acceptor probes are placed at the labeled sites and (**d**) average FRET efficiencies are estimated according to different averaging regimes.

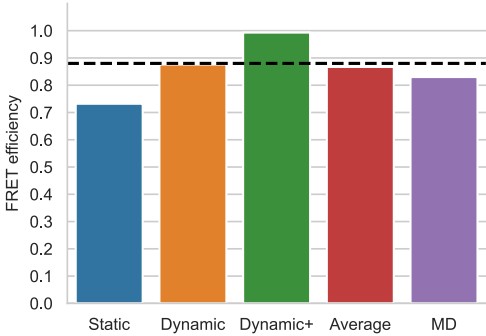

**Fig. 3 | FRET efficiencies for polyproline 11.** FRET efficiencies calculated using FRETpredict and the MD trajectory of polyproline 11 fluorescently labeled at the terminal residues. We calculated $E$ using the large rotamer libraries and for the different regimes (static, dynamic, and dynamic+, in blue, orange, and green, respectively). The graph also shows the average over the three regimes (red) and the $E$ value obtained from MD simulation with explicit FRET probes (purple). The black dashed line indicates the experimental $E$ value.

Line two generates the `MDAnalysis Universe` object from an XTC trajectory and a PDB topology. Line three initializes the FRETpredict object with the labeled residue numbers, the FRET probes from the available rotamer libraries, and turns the electrostatic calculations on. Line seven runs the calculations and saves per-frame and ensemble-averaged data to file. Line eight recomputes the FRET efficiencies using per-frame weights calculated based on dye-protein energies and saved to file by the `FRETpredict.run()` function. $R_0$ was computed for each combination of FRET probes via Eq. (1).

Since pp11 adopts rod-like conformations, steric clashes between rotamers and protein occur in a small fraction of frames (2.7%). Accordingly, applying the reweighting approach (Methods and Supplementary Note 3) to this system leads to similar predictions as the default scheme, i.e., discarding frames with steric clashes and assigning equal weights to the remainder frames (Supplementary Fig. 8).

**Case study 2: conformational ensemble of an intrinsically disordered protein**
ACTR (the activation domain of the activator for thyroid hormone and retinoid receptors) is an intrinsically disordered protein that has previously been extensively studied[44–46]. Here, we used ACTR to demonstrate how FRETpredict can be used on conformational ensembles for intrinsically disordered proteins.

We used previous experimental FRET measurements and MD simulations for ACTR solutions at different urea concentrations that were used to assess the effect of chemical denaturants on protein structure[47,48]. As in the experiments, we labeled the residue pairs 3-61, 3-75, and 33-75 with Alexa Fluor 488 - C1R as the donor and Alexa Fluor 594 - C1R as the acceptor. To account for the dependence of $R_0$ on urea concentration, we used Eq. 4 in Zheng et al.[47] and estimated $R_0 = 5.40$ Å, 5.34 Å, and 5.29 Å for [urea] = 0 M, 2.5 M, and 5 M, respectively.

Figure 4 and Supplementary Table 3 show the FRET efficiency values predicted by FRETpredict at the different urea concentrations (0 M, 2.5 M, and 5 M) using medium rotamer libraries. The absolute values of predicted FRET efficiency differ from the experimental values on average by 13.1%, 7.2%, and 12.1% for [urea] = 0 M, 2.5 M, and 5 M, respectively. Notably, the predicted trend is consistent with the experimental data for all the pairs of labeled residues of ACTR and at the three urea concentrations. The agreement between calculated and experimental trends for the $E$ values shown in Fig. 4 relies on the thorough and accurate sampling of conformational ensembles obtained via MD simulations by Zheng et al.[47] while it also contributes to validating FRETpredict as a model for calculating $E$.

To determine which regime most accurately predicts the FRET efficiency, we calculated the root-mean-square error (RMSE) between the predicted and experimental values for all the residue pairs. For the ACTR data, RMSE values obtained for the static, dynamic, and dynamic+ regimes and their average for all the urea concentrations are 0.233, 0.177, 0.315, and 0.171, respectively. As observed in Case Study 1, the dynamic regime and the average best approximate the experimental FRET efficiency data.

The following lines of code were used to calculate the $E$ values from the ACTR trajectory at [urea] = 0 M:

```
from FRETpredict import FRETpredict
u_0M = MDAnalysis.Universe("actr.gro", "actr_urea0.xtc")
FRET = FRETpredict (protein=u_0M, residues=[3, 61],
        fixed_R0=True, r0=5.40,
        electrostatic=True,
        libname_1="AlexaFluor 488 C1R cutoff20",
        libname_2="AlexaFluor 594 C1R cutoff20")
FRET.run()
FRET.reweight()
```

Line two generates the `MDAnalysis Universe` object from an XTC trajectory and a GRO topology. Line three initializes the FRETpredict object with the labeled residue numbers, the FRET probes from the available rotamer libraries, and fixes the $R_0$ value corresponding to the specific urea concentration listed above. Line eight runs the calculations and saves per-frame and ensemble-averaged data to file. Line nine recomputes the FRET efficiencies using per-frame weights.

Since ACTR adopts more collapsed conformations in pure water than in the presence of urea, we expect the interaction between the protein and the inserted rotamers to be dominated by steric repulsion in a larger fraction of frames at [urea] = 0 M. Indeed, when applying the reweighting approach based on rotamer-protein interactions (Methods and Supplementary Note 3), we estimate that the effective fraction of frames contributing to the reweighted ensemble are 46.9%, 73.0%, and 67.2% for [urea] = 0, 2.5, and 5 M, respectively. Accordingly, the RMSE between FRET efficiencies predicted with and without the reweighting approach is the highest for [urea] = 0 M (Fig. 5 and Supplementary Fig. 9). Thus, reweighting improves agreement with experiments, particularly for the most structured ensemble (we note that the reweighting is based on probe-protein interactions and is not against the experimental data). Here, the accuracy of the underlying ACTR protein ensembles is supported by the good agreement with independent SAXS experiments[47].

**Case study 3: single protein structures**
Although we generated rotamer libraries for several of the most common FRET probes, in some cases smFRET experiments might be performed with probes that are currently not available in FRETpredict. In this case study, we illustrate how, in the absence of the exact probes, accurate trends can be predicted by (i) choosing rotamer libraries with similar structural characteristics (linker length, linker dihedrals, fluorophore structure) and (ii) entering the $R_0$ for the experimental pair of dyes (Supplementary Fig. 10). We apply this strategy to the single structures of HiSiaP, SBD2, and MalE and show that it leads to results that are consistent with the experimental trends. The reference FRET efficiency data of this case study was obtained from experiments by Peter et al.[49], wherein Alexa Fluor 555 - C2R and Alexa Fluor 647 - C2R dyes were employed as donor and acceptor, respectively. In FRETpredict, both donor and acceptor were replaced by AlexaFluor 647 - C2R, the available rotamer library with the most similar steric hindrance (Supplementary Fig. 1), whereas we used the $R_0$ value of the FRET pair used in the actual experiments.

HiSiaP is the periplasmic substrate-binding protein from the sialic acid tripartite ATP-independent periplasmic transporter of *Haemophilus influenzae*. In this protein, ligand binding induces a conformational rearrangement from an open to a closed state[50]. We calculated $E$ values for the labeled residue pairs measured by Peter et al.[49] (58-134, 55-175, 175-228, and

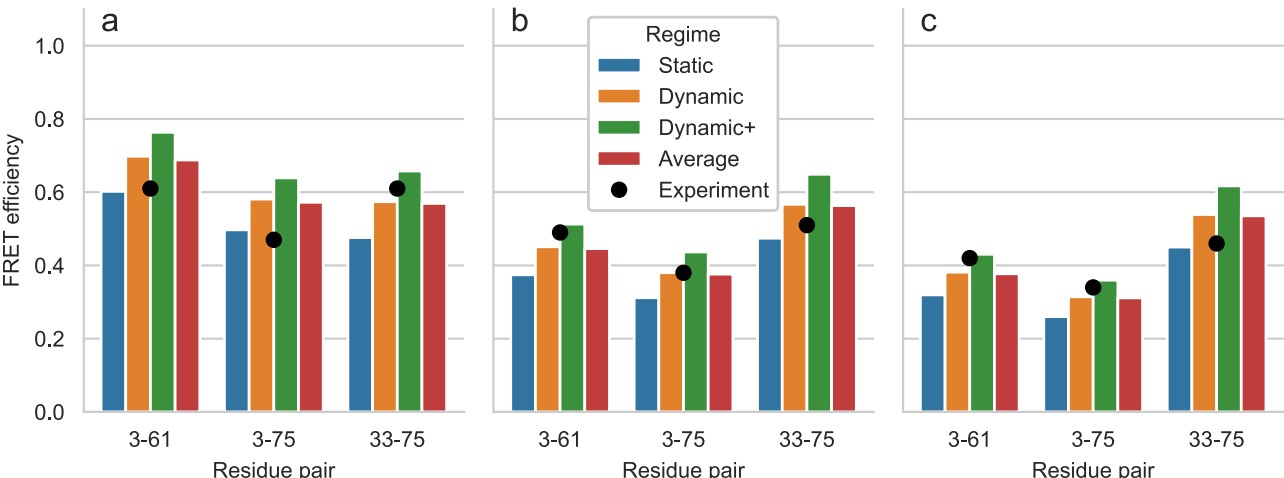

**Fig. 4 | FRET efficiency for ACTR at three urea concentrations.** FRET efficiency for ACTR at [urea] = 0 M (**a**), 2.5 M (**b**), and 5 M (**c**). The protein is fluorescently labeled at three different pairs of sites: 3-61, 3-75, and 33-75. Bars show FRETpredict estimates of the $E$ values calculated using medium rotamer libraries. Predictions for the static, dynamic, and dynamic+ regimes and their average are shown as blue, orange, green, and red bars, respectively. Black circles show the experimental data from Borgia et al.[48].

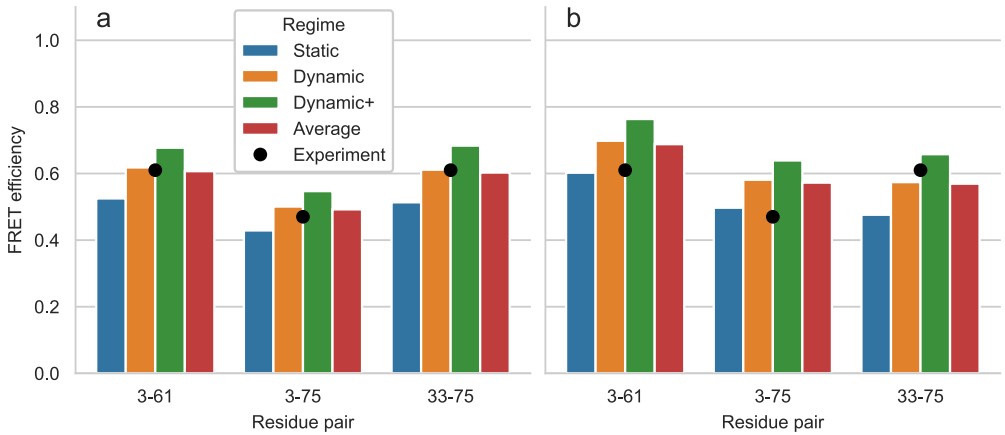

**Fig. 5 | Effect of reweighting on predicted FRET efficiencies.** FRET efficiency for ACTR at [urea] = 0 M (**a**) with and (**b**) without reweighting the trajectory frames based on rotamer-protein interactions. The protein is fluorescently labeled at three different pairs of sites: 3-61, 3-75, and 33-75. Bars show FRETpredict estimates of the $E$ values calculated using medium rotamer libraries. Predictions for the static, dynamic, and dynamic+ regimes and their average are shown as blue, orange, green, and red bars, respectively. Black circles show the experimental data from Borgia et al.[48].

112-175) using structures deposited in the Protein Data Bank (PDB) for the open and closed structures (PDB codes 2CEY[51] and 3B50[52], respectively). The absolute values of the FRET efficiency predicted for HiSiaP differ on average by 20.6% and 24.3% from the experimental values of the open and closed conformation, respectively (Fig. 6a and b, and Supplementary Table 4). The trend of the FRETpredict prediction is about equally consistent with the experimental data for both conformations and for all the pairs of labeled residues. The code used to calculate the FRET efficiency for the single HiSiaP open structure with FRETpredict is:

```
from FRETpredict import FRETpredict
u_open = MDAnalysis.Universe("2cey.pdb")
FRET = FRETpredict (protein=u_open, residues=[58,
134], temperature=298
        fixed_R0=True, r0=5.1,
        electrostatic=True,
        libname_1="AlexaFluor 647 C2R cutoff10",
        libname_2="AlexaFluor 647 C2R cutoff10")
FRET.run()
FRET.reweight ()
```

Line two generates the `MDAnalysis Universe` object for the open structure from a PDB topology. Line three initializes the FRETpredict object with the labeled residue numbers, the FRET probes from the available rotamer libraries, and fixes the $R_0$ value to the experimental one. Line eight runs the calculations and saves per-frame and ensemble-averaged data to file. The same FRETpredict code structure has been used for the other single structure tests of SBD2 and MalE.

SBD2 is the second of two substrate-binding domains constituting the glutamine ABC transporter GlnPQ from *Lactococcus lactis*. As for HiSiaP, upon binding of high-affinity ligands, SBD2 undergoes a transition from an open to a closed state[53]. Peter et al.[49] performed FRET efficiency measurements on SBD2 by labeling the residue pairs 319-392 and 369-451. We used the structures for the open and closed structures deposited in the PDB (PDB codes 4KR5[54] and 4KQP[54], respectively) in combination with AlexaFluor 647 - C2R as both donor and acceptor. The absolute values of the FRET efficiency predicted for SBD2 differ on average by 21.6% and 21.1% from the experimental values of the open and closed conformation, respectively (Fig. 6c and d, and Supplementary Table 4).

The maltose binding protein from *Escherichia coli*, MalE, plays an important role in the uptake of maltose and maltodextrins by the maltose

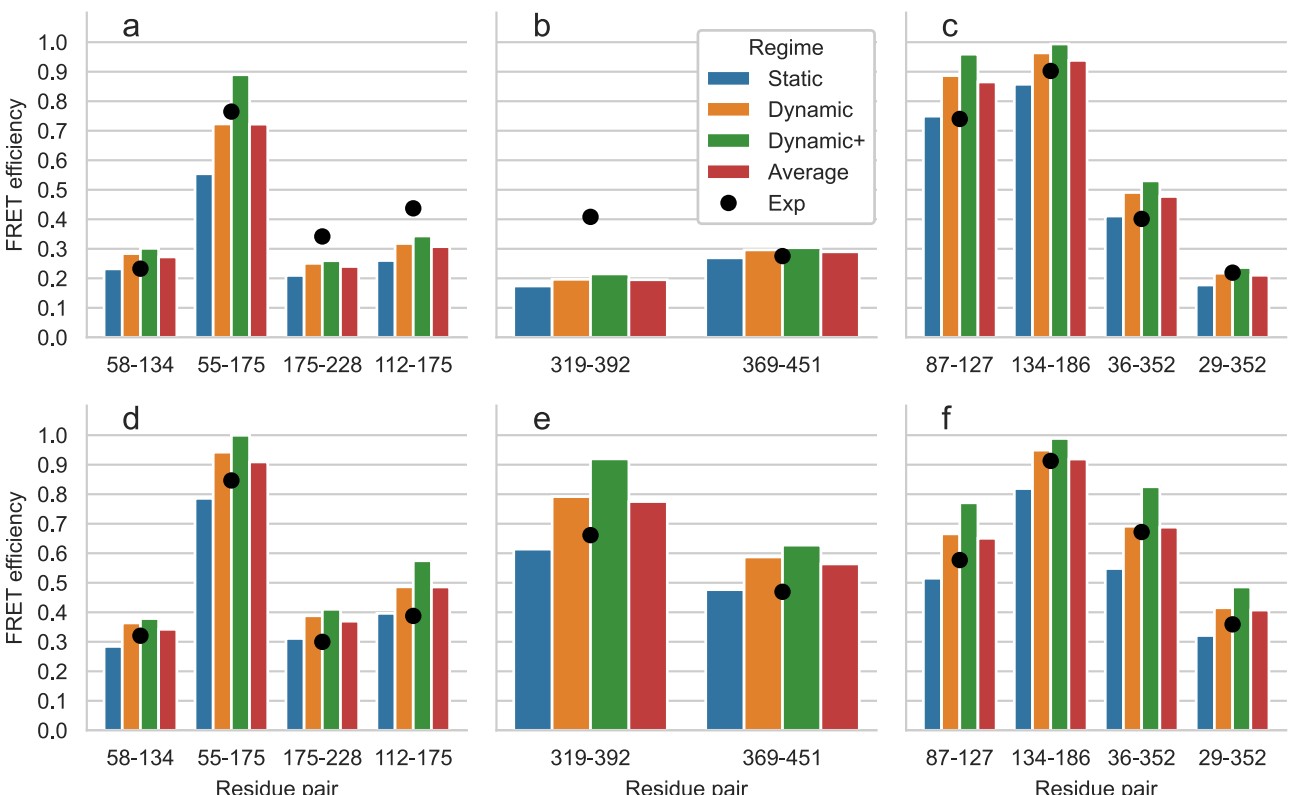

**Fig. 6 | Predictions of FRET efficiencies for single structures.** FRET efficiency values obtained on the single structures for the open (**a–c**) and closed (**d–f**) conformations of HiSiaP (**a** and **d**), SBD2 (**b** and **e**), and MalE (**c** and **f**) for the different residue pairs, using large rotamer libraries. Predictions for the static, dynamic, and dynamic+ regimes and their average are shown as blue, orange, green, and red bars, respectively. Black circles show the experimental reference values from Peter et al.[49] for each pair of residues.

transporter complex MalFGK$_2$[55]. MalE undergoes structural transition between the apo and holo states upon sugar binding, resulting in a ca. 35° rigid body domain reorientation[56]. Peter et al.[49] performed FRET measurements on MalE by labeling the residue pairs 87-127, 134-186, 36-352, and 29-352. We used open and closed structures (PDB codes 1OMP[57] and 1ANF[58], respectively) with AlexaFluor 647 - C2R as both donor and acceptor. The absolute values of the FRET efficiency predicted for MalE differ on average by 15.1% and 10.0% from the experimental values of the open and closed conformation, respectively (Fig. 6e and f, and Supplementary Table 4). The RMSE values associated with the three averaging regimes over all single-frame structures of HiSiaP, SBD2, and MalE are 0.097 (static), 0.094 (dynamic), 0.141 (dynamic+), and 0.086 (average). Based on these results, we observe that even in the case of single-frame structures, the best predictions correspond to the Dynamic regime.

In this case study, we used probes that are similar but not identical to those used in the experiments. The main physicochemical factors to take into consideration to assess the similarity between probes are the steric bulk of the dye, the length and flexibility of the linker, and the presence of charged groups. We already noted that the steric bulk of the FRET probe and the rigidity of the linker have a strong influence on the clustering of the rotamers. Accordingly, these structural features also affect the external weights calculated upon placement of the rotamers at the binding site. On the other hand, we observed that including electrostatic interactions in FRETpredict calculations (`electrostatic=true`) had little effect on the accuracy of FRET efficiency predictions for the studied systems (Supplementary Fig. 10). In summary, we found that using the rotamer library for a probe with similar steric hindrance, in combination with the $R_0$ value for the correct dye pair, yields FRET efficiency trends in good agreement with the experimental data (Supplementary Fig. 10).

## Discussion

We have introduced FRETpredict, a Python-based open-source software with a fast implementation of the RLA for the calculation of FRET efficiency data. FRETpredict's primary purpose is to be a tool that is easy to use—also for large conformational ensembles—and we provide access to rotamer libraries for many dyes and linkers. Users can also use their own generated libraries following the procedure detailed above. The main advance of this work is not the use of the rotamer library approach for FRET calculations (already described for example by Walczewska-Szewc et al.[22] and Klose et al.[33]) but rather how FRETpredict enables researchers to use such libraries more easily and for a wider range of problems.

Using three case studies, we have highlighted the capabilities of our implementation in the case of a peptide trajectory (pp11), an IDP trajectory (ACTR), and single protein structures (HiSiaP, SBD2, and MalE). The FRET efficiency prediction trends are in most cases in good agreement with the experimental data. However, we note that the accuracy of the method depends on the quality and relevance of the protein conformational ensembles that are used as input.

In FRETpredict, the average FRET efficiency can be calculated in three different regimes: static, dynamic, and dynamic+. The Dynamic regime has been shown to provide better agreement with experimental data for both protein conformational ensembles and single protein structures. In the absence of information about the different timescales, we find that simply averaging the results from the three regimes often leads to good agreement with experiments. The averaging over conformational ensembles in the various regimes can also be performed by assigning weights to each protein conformation based on the interaction energy between dyes and protein. In our case studies, this reweighting approach can result in better predictions compared to excluding frames with steric clashes, but its accuracy and general utility need further validation.

FRETpredict calculations and, more generally, FRET efficiency predictions from protein trajectories involve a trade-off between computation time and prediction accuracy. Accordingly, the choice of the optimal rotamer library selection must take its size into consideration. Large rotamer libraries may lead to greater accuracy but are also more computationally expensive than smaller libraries. On the other hand, both medium and small rotamer libraries are a good compromise between calculation time and accuracy when long simulation trajectories are used. However, using a small number of rotamer clusters (i.e., small rotamer libraries) may compromise the prediction of FRET efficiency, especially in case of tight placement at the labeled site, in which many rotamers may be excluded from the calculation due to probe-protein steric clashes. Therefore, we recommend using large rotamer libraries when the computational cost is not a limiting factor and medium libraries for larger conformational ensembles.

Possible application scenarios include coupling FRETpredict more directly with methods that generate structures in a 'modeling loop', i.e. improving a model by minimizing the difference between prediction and experimental values. It is also possible to benchmark simulations, test or rank structural models, optimize force fields against FRET data, or generate input to so-called reweighting approaches (as has also been done using EPR data[59]).

FRETpredict has a general framework and can be readily extended to encompass non-protein biomolecules and additional rotamer libraries of FRET probes. In the current implementation, we consider all combinations of rotamers from the respective donor and acceptor libraries and independently weigh each dye based on protein-dye interaction energies, which are evaluated for the two rotamers independently. The approach could be further developed to randomly sample pairs of rotamers and to account for dye-dye interactions in the calculation of the statistical weights assigned to each pair. Further, the calculation of average FRET efficiencies could be based on the diffusive motion of the FRET probes in a potential of mean force derived from donor–acceptor distance distributions, as recently described[60] and implemented in the MMM software-tool[33].

## Methods
### FRET efficiency calculation
FRET efficiency is defined as the fraction of donor excitations that result in energy transfer to the acceptor, and can be calculated as $E = \frac{k_{ET}}{k_D + k_{ET}}$, where $k_{ET}$ is the instantaneous FRET rate and $k_D$ is the spontaneous decay rate of donor excitation by non-FRET mechanisms (e.g. donor emission or non-radiative mechanisms). $k_{ET}$ can be calculated as $k_{ET}(\kappa^2, r) = \frac{3}{2} k_D \kappa^2 \left(\frac{R_0}{r}\right)^6$, where $R_0$ is the Förster radius, and $\kappa^2$ is the orientation factor, related to the relative orientation of the dipole moments of the dyes. The Förster radius is defined as

$$R_0 = 0.02108 \left(J \kappa^2 Q_D n^{-4}\right)^{1/6} \text{ Å}, \tag{1}$$

where $J$ is the spectral overlap integral between the fluorescence emission of the donor and the absorption spectrum of the acceptor, $Q_D$ is the quantum yield of the donor in the absence of the acceptor, and $n$ is the refractive index of the medium. Of these parameters, the most challenging to estimate is $\kappa^2$. While it can be difficult to measure $\kappa^2$ experimentally due to the rapid isomerization of the linker region of the probes, $\kappa^2$ is often approximated to its freely diffusing isotropic average of 2/3 by considering that the fluorophore dynamics occur on a timescale that is sufficiently shorter than the donor lifetime. By assuming a fixed donor–acceptor distance, $r$, and $\kappa^2 = 2/3$, we obtain

$$E = \frac{R_0^6}{r^6 + R_0^6}. \tag{2}$$

For most cases, this approximation is acceptable due to the length of the linker region and rapid fluorophore reorientation. However, the placement of the probes on a protein structure may restrict the motions of the dyes due to interactions with the surrounding protein environment. Because of such potentially restricted fluorophore motions, sometimes $\kappa^2 \neq 2/3$. Therefore, a more general formula for calculating FRET efficiency is

$$E(r, \kappa^2) = \left(1 + \frac{2}{3\kappa^2} \left(\frac{r}{R_0}\right)^6\right)^{-1}. \tag{3}$$

In this case, it is still assumed that the chromophore is reorienting faster than the donor lifetime, but that its motion is restricted in space. Due to the discrete nature of the RLA, FRETpredict allows precise computation of $\kappa^2$ and the possibility to compute $R_0$ in a $\kappa^2$-dependent way. $\kappa^2$-dependent $R_0$ calculations (Eq. (1)) are the default in FRETpredict, but users can also adopt a fixed $R_0$ value by setting `fixed_R0=True` and specifying the $R_0$ value with the `r0` option. $R_0$ values for the most common FRET probes are reported in *lib/R0/R0_pairs.csv*.

### Averaging regimes
Protein, linker and dye motions may all contribute to FRET and so dynamics on different timescales may be important; here we simplify these as the protein correlation time ($\tau_p$), the linker-distance correlation time ($\tau_l$), the orientation correlation time of the dye ($\tau_k$), and the fluorescence lifetime ($\tau_f$). Given a conformational ensemble, but no explicit representation of the dynamical motion and timescales, the "average" FRET efficiency depends on how rapidly the various time-dependent components of $E$ (i.e., $r$ and $\kappa^2$ in Eq. (3)) are averaged relative to the fluorescence lifetime. If a specific motion occurs much faster than the fluorescence decay, the effective $k_{ET}$ will be completely averaged over that degree of freedom. Assuming that protein fluctuations are slow (i.e., $\tau_p >> \tau_f$), we obtain three different regimes for the relationship between the experimentally measured efficiency and the underlying donor–acceptor distance distribution[39].

In the static regime ($\tau_k >> \tau_f$ and $\tau_l >> \tau_f$), dye distance and orientation fluctuations are both slow, thus, there is no averaging of transfer rate, and every combination of protein configurations, linker distance, and dye orientation gives a separate $k_{ET}$. In this case, the FRET efficiency is averaged over $N$ protein conformations as well as over the $m$ and $l$ rotamers for the donor and the acceptor, respectively,

$$\langle E \rangle_{static} = \frac{1}{N} \sum_{s=0}^{N} \sum_{j=0}^{m} \sum_{i=0}^{l} \left(1 + \frac{2}{3\kappa_{sij}^2} \left(\frac{r_{sij}}{R_0}\right)^6\right)^{-1} \times p_{si} \times p_{sj}. \tag{4}$$

where $p_{si}$ and $p_{sj}$ are weights corresponding to the rotamers $i$ and $j$ of the fluorophores in conformation $s$. In this regime, $\kappa_{sij}^2$ is an instantaneous value calculated for a given combination of donor and acceptor rotamers as

$$\kappa_{sij}^2 = \left(\hat{\mu}_i \cdot \hat{\mu}_j - 3\left(\hat{R}_{sij} \cdot \hat{\mu}_j\right)\left(\hat{R}_{sij} \cdot \hat{\mu}_i\right)\right)^2, \tag{5}$$

where $\hat{\mu}_{si}$ and $\hat{\mu}_{sj}$ are the transition dipole moment unit vectors of the donor and acceptor, respectively, and $\hat{R}_{sij}$ denotes the normalized inter-fluorophore displacement. In FRETpredict, the atom pairs defining $\hat{\mu}_{si}$, $\hat{\mu}_{sj}$, and $\hat{R}_{sij}$ are specified in *lib/libraries.yml*.

In the dynamic regime ($\tau_k << \tau_f$ and $\tau_l >> \tau_f$) the complete conformational sampling is achieved within the fluorescence lifetime of the donor. In this scenario, which is commonly assumed in the treatment of experimental data, the FRET efficiency is calculated as

$$\langle E \rangle_{dynamic} = \frac{1}{N} \sum_{s=0}^{N} \sum_{j=0}^{m} \sum_{i=0}^{l} \left(1 + \frac{2}{3\langle \kappa^2 \rangle} \left(\frac{r_{sij}}{R_0}\right)^6\right)^{-1} \times p_{si} \times p_{sj}. \tag{6}$$

Here, $\langle \kappa^2 \rangle$ is calculated over all the protein conformations and combinations of probe rotamers:

$$\langle \kappa^2 \rangle = \frac{1}{N} \sum_{s=0}^{N} \sum_{j=0}^{m} \sum_{i=0}^{l} \kappa_{sij}^2 \times p_{si} \times p_{sj}, \qquad (7)$$

In the dynamic+ regime ($\tau_k << \tau_f$ and $\tau_l << \tau_f$), both dye distances and orientations are very fast, and the $k_{ET}$ for each protein frame is averaged over all dye configurations, considering both distances and orientations. The FRET efficiency is calculated as

$$\langle E \rangle_{dynamic+} = \frac{1}{N} \sum_{s=0}^{N} \frac{A_s}{1+A_s}, \qquad (8)$$

where

$$A_s = \sum_{j=0}^{m} \sum_{i=0}^{l} \frac{3}{2} \kappa_{sij}^2 \left( \frac{R_0}{r_{sij}} \right)^6 \times p_{si} \times p_{sj}. \qquad (9)$$

### Reweighting

In all averaging regimes, protein conformations with steric clashes are discarded and the remainder are weighted equally. Besides this default scheme, the code allows for reweighting the frames based on dye-protein interactions (Supplementary Note 3). This approach is implemented in the `FRETpredict.reweight()` function, which efficiently recomputes $\langle E \rangle$ using the pre-calculated per-frame values. Moreover, through the custom parameter `user_weights`, the per-frame weights calculated internally can be combined with user-provided statistical weights from other methods, such as Bayesian/Maximum Entropy[61] or enhanced sampling techniques[62].

### Statistics and reproducibility

No statistical methods were used to predetermine sample sizes before analyzing the data presented in this study. No data were excluded from the analysis. To facilitate the reproducibility of our results, we make available code and data on GitHub and Zenodo.

### Reporting summary

Further information on research design is available in the Nature Portfolio Reporting Summary linked to this article.

### Data availability

Data underlying the analyses and figures presented in this work are available at github.com/KULL-Centre/_2023_Montepietra_FRET and on Zenodo at https://doi.org/10.5281/zenodo.10573638[63]. Simulation trajectories of the dyes in water are available upon request.

### Code availability

The FRETpredict package is available on GitHub at github.com/KULL-Centre/FRETpredict and on Zenodo at https://doi.org/10.5281/zenodo.10371378[64]. The repository includes tutorials for predicting FRET efficiencies and creating new rotamer libraries. FRETpredict is licensed under GPL license v3 and also distributed as a PyPI package (pypi.org/project/FRETpredict). Jupyter notebooks to reproduce the analyses and figures presented in this work are available at github.com/KULL-Centre/_2023_Montepietra_FRET and on Zenodo at https://doi.org/10.5281/zenodo.10573638[63].

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

## Acknowledgements
M.B.A.K. acknowledges funding from the Lundbeck Foundation (lundbeckfonden.com). K.L.-L. acknowledges funding via a Sapere Aude Starting Grant from the Danish Council for Independent Research (Natur og Univers, Det Frie Forskningsråd, 12-126214, https://dff.dk/) and the Lundbeck Foundation BRAINSTRUC initiative in structural biology (R155-2015-2666, lundbeckfonden.com). We acknowledge the use of resources at the core facility for biocomputing at the Department of Biology. R.B.B. was supported by the Intramural Research Program of the National Institute of Diabetes and Digestive and Kidney Diseases of the National Institutes of Health. This work utilized the computational resources of the NIH HPC Biowulf cluster. (http://hpc.nih.gov).

## Author contributions
D.M., G.T., J.M.M., M.B.A.K., R.B.B., and K.L.-L. designed research; D.M., G.T., J.M.M., M.B.A.K., and R.B.B. performed research; D.M., G.T., J.M.M., M.B.A.K., and R.B.B. analyzed data; D.M., G.T., R.B.B., and K.L.-L. wrote the paper; K.L.-L. supervised the study.

## Competing interests
K.L.-L. holds stock options in and is a consultant for Peptone Ltd. All other authors declare no competing interests.
