## [Peer Review File · Communications Biology]

Reviewers' comments:

Reviewer #1 (Remarks to the Author):

This manuscript introduces a general approach for predicting FRET efficiencies from MD trajectories of unlabeled proteins or other molecules or from sufficiently detailed ensemble models. Given the broad applicability and wide use of FRET in biological studies, such work is of interest for Communications Biology. The approach strikes a nice balance between computational effort and accuracy and I expect it to be very useful in future research. The authors have implemented some flexibility for fine tuning the algorithm. They give clear recommendations for current best practice in such fine tuning, based on comparison for different types of proteins between rotamer library prediction and more elaborate MD prediction or between rotamer library prediction and experiment. These tests indicate useful accuracy of the approach. The manuscript is mostly clear and, together with Supporting Information, describes the work in sufficient detail. A few points and a few typos require minor revision.

1. The Results section starts with a discussion of Figure 2 and rotamer library size reduction, without giving at least rough information how large the respective libraries are. Please mention in the caption of Figure 2 what is meant by “large” and in the main text how large the libraries are roughly after clustering/filtering. In this, you might want to refer to Figure S10.

2. Although it is a nice observation that some predictions can be made with rotamer libraries of similar chromophores instead of the actually used ones, I would usually prefer to use libraries for the correct chromophores. Please provide an estimate of the computation time required for library generation. How difficult would it be to parameterize the force field for a new chromophore?

3. The polyproline case study appears to assume that more elaborate MD simulations are kind of a gold standard. This assumption did not hold in similar studies for spin labels, where agreement with experiment was not consistently better for MD simulations than for rotamer library predictions. The reason is probably that sampling of feasible linker conformations for a given protein backbone conformation is better with a rotamer library than with standard MD of the labelled protein. I recommend to make a cautionary remark.

4. The manuscript is somewhat deficient in discussing application scenarios for the approach. It is clear that the approach could reveal wrong models by predicting FRET efficiencies that contradict experimental results. But would it also be possible to use this approach in a “modelling loop”, i.e. for improving a model by minimizing difference between prediction and experimental values? This is done with spin labels and DEER measurements. It would be nice to see how the authors assess the potential for chromophores and FRET.

Typos:

p. 8: “hence to its the rotational degrees” should read “hence to its rotational degrees”

p. 9 “steric clashed between rotamers” should read “steric clashes between rotamers”

SI p. 1 (p. 25 in PDF): “conformational ensembles frem REMD as input” should read “conformational ensembles from REMD as input”

SI Table S11: the entries for “small” and “large” should be interchanged

Reviewer #2 (Remarks to the Author):

The manuscript describes a very useful tool to predict FRET efficiency using one or many protein structures using rotamer libraries. This accounts for the size and mobility of fluorophore linkers to yield a more accurate prediction of the expected FRET from a given structure than would be the case without modelling in the likely dye positions. The manuscript is well written and very clear to read. The software is clear and easy to use and is likely to be useful to a large number of scientists conducting FRET experiments. One concern I have with the current manuscript is that very similar software has previously been described that first introduced the rotamer library approach (RLA) to FRET (ref 29), along with a set of very similar comparisons to experimental data and very similar experimental situations. I feel like this similarity is rather overlooked here and reduces the novelty of the current work. That said, while very similar in implementation, the current software is easier to use, has a larger rotamer library set and authors that are currently more active in the field to keep it up to date for FRET experimentalists. As such it is likely to be used in the research field.

There is also very similar software for available volume analysis of MD protein trajectories that should be referenced for comparison (<https://doi.org/10.1093/bioinformatics/btab615>)

In the 4th paragraph of the introduction, the authors discuss various approached to model the conformational space of dyes attached to proteins including available volume, full MD and RLA. While the time taken for these is mentioned, a discussion of the relative accuracy of each is important here to justify the use of the RLA. If this is not done here, please cite previous studies that do. A downside of full MD is suggested to be the need for force field parameterization of the fluorescent dyes, but this is also required to make rotamer libraries so this should be clarified. In addition, MD simulations need not necessarily provide the best result unless they are able to adequately sample the full range of dye conformations for a given protein conformation. There are some studies of how well this can be done that could be mentioned here (eg <https://doi.org/10.1021/acs.jctc.5b00205> but I think there are others)., The discussion makes a very good point about MD reproducing conformational states of the protein, but even reproducing conformations of the dyes for a static protein conformation is not easy!

A nice addition to the method in this paper is the ability to reweight the rotamer probabilities based upon the interaction energies with the surrounding protein. I really like this concept, but currently there

is no data to validate that this approach is better than not weighting the probabilities. Such reweighting will always bias rotamers that lie near the protein as they are not balanced by interactions with surrounding water (it is effectively reweighting in the absence of solvent). Thus, it is possible such reweighting makes the distribution worse not better. It is great that this is optional. Currently the only comparison of the two approaches is against experimental data (Fig 5) but here we don't know if the underlying ensemble of protein conformations is influencing the outcome. A better justification of the approach would be to compare against a well converged full MD simulation of the dyes for a static protein. This is a significant addition to the paper but would provide much better justification for the approach. If this is not done then I would recommend removing the claim conclusion that reweighting is more accurate as currently there is not data to support this.

I would like a brief discussion of how the rotamers are filtered and clustered to be included in the main text – it can be just a couple of sentences but this is important to the method. Also, the concepts of small medium and large libraries are not introduced before they are used in the results. This should be defined and ideally better supporting data for the choice of library size and how this influences accuracy should be provided or referenced.

The different averaging regimes are very well described and implemented in the code – the authors should be congratulated on this. In the conclusion it is stated that they recommend the static regime for single structures. The reason for this is not clear to me. If the dynamics regime is better when you have an ensemble of slowly moving protein structures, wont this also be the best for a single protein structure?

Minor points:

The introduction starts by describing single molecule FRET experiments and the use of rotamer library models for these. In fact, it is just as useful for ensemble FRET measurements so I would recommend noting this.

The references to prior direct MD simulations of FRET dyes attached to biomolecules are somewhat selective to those by the authors. There are many extensive studies by the labs of Grubmuller, Corry and others that could be mentioned. Perhaps this need not be exhaustive in the introduction but only citing those by the current authors is not ideal.

The current software outputs the average FRET efficiency as plotted in the nice bar graphs. For smFRET experiments it would also be useful to display the FRET efficiency distributions that can directly be compared to the raw experimental data.

Reviewer #3 (Remarks to the Author):

Montepietra and coworkers introduced a package called FRETpredict, which is able to predict FRET efficiencies from ensembles of protein conformations. As the paper shows that it can utilize a rotamer library approach to describe the FRET probes covalently bound to the protein. And they have tested on different types of systems: a relatively structured peptide and three folded proteins (HiSiaP, SBD2, and MalE). Overall, the paper was presented in a clear format with methodology and case studies in each section. The github and tutorials are also well documented. I feel the statistics and software will benefit a lot to this field of prediction and would recommend publication with minor concerns as follows:

1. Looks like all the efficiency plots do not have an error bar, does it mean the plots are obtained within 1-time trial?

2. The folded proteins are among transporters which means more membrane protein and the protein size looks not that big.

Can the author explain why to chose them as case study proteins, and are they the standard protein designed as test cases for FRET prediction? Will larger membrane protein work for FRETpredict?

also, will the authors try some soluble and important proteins as kinase for the test cases too?

Reviewer 1

This manuscript introduces a general approach for predicting FRET efficiencies from MD trajectories of unlabeled proteins or other molecules or from sufficiently detailed ensemble models. Given the broad applicability and wide use of FRET in biological studies, such work is of interest for Communications Biology. The approach strikes a nice balance between computational effort and accuracy and I expect it to be very useful in future research. The authors have implemented some flexibility for fine tuning the algorithm. They give clear recommendations for current best practice in such fine tuning, based on comparison for different types of proteins between rotamer library prediction and more elaborate MD prediction or between rotamer library prediction and experiment. These tests indicate useful accuracy of the approach. The manuscript is mostly clear and, together with Supporting Information, describes the work in sufficient detail. A few points and a few typos require minor revision.

We are happy that the reviewer found our work interesting and useful and thank them for their constructive comments.

1. The Results section starts with a discussion of Figure 2 and rotamer library size reduction, without giving at least rough information how large the respective libraries are. Please mention in the caption of Figure 2 what is meant by “large” and in the main text how large the libraries are roughly after clustering/filtering. In this, you might want to refer to Figure S10.

In the revised “Rotamer library Generation” subsection in the main text, we now outline the procedure used to generate the rotamer libraries:

“In FRETpredict, rotamer libraries are created through the following steps: (i) generation of the conformational ensemble of the FRET probe using MD simulations; (ii) selection of combinations of the most populated dihedral angles to generate the C1 set of cluster centers; (iii) assignment of trajectory frames to the C1 set based on the least-square deviations of the dihedral angles; (iv) average over the angles of the trajectory frames of each C1 cluster center to generate the C2 set of cluster centers; (v) assignment of trajectory frames to the C2 set based on the least-square deviations of the dihedral angles; (vi) filtering of clusters with populations lower than 10, 20, and 30 structures to generate the rotamer libraries referred to as large, medium, and small hereafter. These steps are detailed in S1 Text and implemented in FRETpredict/rotamer_libraries.py.”

In the revised manuscript, we now describe the difference between large, medium, and small rotamer libraries at the beginning of the Results section:

“As detailed in Supporting Text 1, we generated rotamer libraries through a series of clustering steps, starting from large conformational ensembles from molecular simulations of the fluorophores in aqueous solutions. To further decrease the size of the rotamer libraries, we filtered out low-populated cluster centers based on three different cutoffs to obtain large, medium, and small rotamer libraries. To illustrate the extent to which the conformational ensemble of the probes is reduced

upon the generation of the rotamer libraries, we plotted the projection on the xy-plane of the distance vectors between the Ca atom and the central atom of the fluorophore (Fig 2 and S4, S5, and S6) of all the generated rotamer libraries (Figs. S1, S2, and S3).”

Finally, in the caption of Fig 2, we give an estimate of the number of structures in the large rotamer libraries and refer to Fig. S7:

“2D projections of the position of the fluorophore with respect to the Ca atom for the large rotamer libraries generated in this work, which typically contain hundreds of structures (Fig. S7).”

2. Although it is a nice observation that some predictions can be made with rotamer libraries of similar chromophores instead of the actually used ones, I would usually prefer to use libraries for the correct chromophores. Please provide an estimate of the computation time required for library generation. How difficult would it be to parameterize the force field for a new chromophore?

We completely agree with the reviewer that it is preferable to use the correct dyes for a calculation; we only provide this option and mention it in the paper to make it clear that people have this option, and that it is also possible (in the computer) to change the spectral properties independently of the chemistry.

Regarding the efforts involved in generating new rotamer libraries, we provide code to generate new libraries as part of FRETpredict (i.e., the Jupyter notebook in tests/tutorials/Tutorial_generate_new_rotamer_libraries.ipynb). There are only a few variables that need to be set (mostly to indicate the relevant torsions in the linker), so the major bottleneck is that one needs to run a molecular dynamics simulation. Here, the limiting factor is whether there is a force field available for the dye and linker, whether one will simply rely on automated tools for force field generation or whether one spends additional time on parameterizing the force field. These issues are now discussed in the revised paper.

3. The polyproline case study appears to assume that more elaborate MD simulations are kind of a gold standard. This assumption did not hold in similar studies for spin labels, where agreement with experiment was not consistently better for MD simulations than for rotamer library predictions. The reason is probably that sampling of feasible linker conformations for a given protein backbone conformation is better with a rotamer library than with standard MD of the labelled protein. I recommend to make a cautionary remark.

We agree with the reviewer that MD simulations with explicit probes may not always provide the most accurate result, and have updated the “Case study 1” paragraph:

“Comparison with the reference values (Fig. 3 and Table S2) indicates that FRETpredict yields predictions that are in slightly better agreement with experiments than MD simulations with explicit representation of the probes. This result suggests that the RLA provides relatively accurate FRET predictions and that MD simulations with explicit probes may not necessarily yield the most accurate

result unless they are able to adequately sample the full range of dye conformations [44, 45].”

We note that in the context of FRET, MD simulations with explicit representations of the probes also make it possible to calculate FRET efficiencies taking the dynamics into account as directly provided by the simulations (something that cannot be done by a rotamer library), though there is of course no guarantee that the simulations provide the correct rates (or ensembles). However, we would like to point out that the MD simulations of polyproline used a force field that had been carefully parameterized, and validated, against experimental data.

4. The manuscript is somewhat deficient in discussing application scenarios for the approach. It is clear that the approach could reveal wrong models by predicting FRET efficiencies that contradict experimental results. But would it also be possible to use this approach in a “modelling loop”, i.e. for improving a model by minimizing difference between prediction and experimental values? This is done with spin labels and DEER measurements. It would be nice to see how the authors assess the potential for chromophores and FRET.

In the revised manuscript, we now discuss potential applications of FRETpredict. In the Conclusion section:

“Possible application scenarios include coupling FRETpredict more directly with methods that generate structures in a “modeling loop”, i.e. improving a model by minimizing the difference between prediction and experimental values. It is also possible to benchmark simulations, test or rank structural models, optimize force fields against FRET data, or generate input to so-called reweighting approaches (as has also been done using EPR data [Kofinger et al. 2019]).”

Typos:

p. 8: “hence to its the rotational degrees” should read “hence to its rotational degrees”

p. 9 “steric clashed between rotamers” should read “steric clashes between rotamers”

SI p. 1 (p. 25 in PDF): “conformational ensembles frem REMD as input” should read “conformational ensembles from REMD as input”

SI Table S11: the entries for “small” and “large” should be interchanged

Thank you; these have now been corrected in the revised manuscript.

Reviewer 2

The manuscript describes a very useful tool to predict FRET efficiency using one or many protein structures using rotamer libraries. This accounts for the size and mobility of fluorophore linkers to yield a more accurate prediction of the expected FRET from a given structure than would be the case without modelling in the likely dye positions. The manuscript is well written and very clear to read. The software is clear and easy to use and is likely to be useful to a large number of scientists conducting FRET experiments. One concern I have with the current manuscript is that very similar software has previously been described that first introduced the rotamer library approach (RLA) to FRET (ref 29), along with a set of very similar comparisons to experimental data and very similar experimental situations. I feel like this similarity is rather overlooked here and reduces the novelty of the current work. That said, while very similar in implementation, the current software is easier to use, has a larger rotamer library set and authors that are currently more active in the field to keep it up to date for FRET experimentalists. As such it is likely to be used in the research field.

We are happy that the reviewer found our work interesting and useful and thank them for their constructive comments. We apologize if the reviewer found that we underplayed the role of the work described in reference 29; this was clearly not our intention. We believe (as the reviewer also states) that there are clear differences. Most centrally, we developed FRETpredict to be a tool that is easy to use—also for large conformational ensembles—and we provide access to rotamer libraries for a relatively large number of dyes and linkers. There are also some differences in implementation, and we believe that the flexibility and ease of use of FRETpredict are important aspects. From reading the reviewer's comments, we believe (and hope) that they agree. In the revised manuscript we have clarified these matters including highlighting that the main step forward in FRETpredict is not the use of rotamer libraries, but rather how FRETpredict enables researchers to use such libraries more easily and for a wider range of problems.

We modified the last paragraph in the Introduction:

“In this work we introduce FRETpredict, an easy-to-use Python module based on the RLA that enables FRET efficiency calculations on a wide range of protein conformational ensembles and MD trajectories. We describe a general methodology to generate rotamer libraries for FRET probes and provide access to rotamer libraries for many commonly used dyes and linkers.”

And added a paragraph in the Conclusions:

“We have introduced FRETpredict, a Python-based open-source software program with a fast implementation of the RLA for the calculation of FRET efficiency data. FRETpredict's primary purpose is to be a tool that is easy to use—also for large conformational ensembles—and we provide access to rotamer libraries for many dyes and linkers. Users can also use their own generated libraries following the procedure detailed above. The main step forward in FRETpredict is not the use of

the rotamer library approach for FRET calculations (already described by Klose et al. and Walczewska-Szewc et al.) but rather how FRETpredict enables researchers to use such libraries more easily and for a wider range of problems.”

We also updated the Abstract to highlight these points:

“Here, we introduce FRETpredict, an easy-to-use Python software program to predict FRET efficiencies from ensembles of protein conformations. FRETpredict uses a rotamer library approach to describe the FRET probes covalently bound to the protein. The software efficiently and flexibly operates on large conformational ensembles such as those generated by molecular dynamics simulations to facilitate the validation or refinement of molecular models and the interpretation of experimental data. We provide access to rotamer libraries for many commonly used dyes and linkers and describe a general methodology to generate new rotamer libraries for FRET probes. We demonstrate the performance and accuracy of the software for different types of systems: a relatively structured peptide (polyproline 11), an intrinsically disordered protein (ACTR), and three folded proteins (HiSiaP, SBD2, and MalE). FRETpredict is open source (GPLv3) and is available at github.com/KULL-Centre/FRETpredict and as a Python PyPI package at pypi.org/project/FRETpredict.”

There is also very similar software for available volume analysis of MD protein trajectories that should be referenced for comparison (<https://doi.org/10.1093/bioinformatics/btab615>)

We thank the reviewer for pointing us to this work, which we now also cite and discuss in the context of the previous text for available volume (AV) calculations.

In the 4th paragraph of the introduction, the authors discuss various approaches to model the conformational space of dyes attached to proteins including available volume, full MD and RLA. While the time taken for these is mentioned, a discussion of the relative accuracy of each is important here to justify the use of the RLA. If this is not done here, please cite previous studies that do. A downside of full MD is suggested to be the need for force field parameterization of the fluorescent dyes, but this is also required to make rotamer libraries so this should be clarified. In addition, MD simulations need not necessarily provide the best result unless they are able to adequately sample the full range of dye conformations for a given protein conformation. There are some studies of how well this can be done that could be mentioned here (eg <https://doi.org/10.1021/acs.jctc.5b00205> but I think there are others)., The discussion makes a very good point about MD reproducing conformational states of the protein, but even reproducing conformations of the dyes for a static protein conformation is not easy!

We agree fully with the comments from the reviewer and have updated the manuscript accordingly.

As for the relative accuracy of different approaches, we agree that these are important issues to address. We, however, also believe benchmark studies should

ideally be published separately from descriptions of new methods/software (see e.g. point 7 of Peters et al, <https://doi.org/10.1371/journal.pcbi.1006494>). We are not aware of previous broad benchmarks, but have added references to previous work comparing e.g. MD simulations or AV calculations with FRET experiments in the 4th paragraph of the Introduction.

As for the computational overhead of MD, it is correct that it also takes some work to generate rotamer libraries (see also answer to Reviewer 1); however, such calculations only need to be performed once and the resulting library can then be applied to many different systems. Further, the simulated system is substantially smaller.

We revised the Introduction to read:

“It also often needs to be preceded by an ad-hoc system set up with fluorescent dyes fully parameterized to be compatible with the force field used.”

“The advantage of the RLA over MD simulations with explicit FRET probes is that it reduces the computational effort, since the calculations required to generate a rotamer library for a new FRET probe only need to be performed once, and the resulting library can then be applied to many different systems. In addition, the simulated system is significantly smaller.”

We also agree that even with very good force fields for proteins and dyes, sampling the full conformational space (in particular for highly flexible dyes) can be a substantial undertaking. We thank the reviewer for pointing us to the work by Walczewska-Szewc et al. and now cite that and related work in the revised manuscript.

A nice addition to the method in this paper is the ability to reweight the rotamer probabilities based upon the interaction energies with the surrounding protein. I really like this concept, but currently there is no data to validate that this approach is better than not weighting the probabilities. Such reweighting will always bias rotamers that lie near the protein as they are not balanced by interactions with surrounding water (it is effectively reweighting in the absence of solvent). Thus, it is possible such reweighting makes the distribution worse not better. It is great that this is optional. Currently the only comparison of the two approaches is against experimental data (Fig 5) but here we don't know if the underlying ensemble of protein conformations is influencing the outcome. A better justification of the approach would be to compare against a well converged full MD simulation of the dyes for a static protein. This is a significant addition to the paper but would provide much better justification for the approach. If this is not done then I would recommend removing the claim conclusion that reweighting is more accurate as currently there is not data to support this.

We thank the reviewer for the comments on our reweighting approach, which indeed is one of the differences between our work and previous work. The reviewer also makes an important point about how to validate such methods. This is in

general very difficult because—as the reviewer notes—one rarely knows the true conformational ensemble, and hence it is difficult to compare different methods for calculating observables.

In the case that we analyse in the paper (ACTR; Figures 5 and S9) we find that reweighting improves agreement with experiments, in particular for the most structured ensemble. Here, we note that these ensembles have also been shown to be in good agreement with SAXS experiments, lending some independent support for the accuracy of the ensembles. We now mention this in the revised paper.

“Thus, reweighting improves agreement with experiments, particularly for the most structured ensemble. Here, the accuracy of the underlying ACTR protein ensembles is supported by the good agreement with independent SAXS experiments performed by Zheng et al.”

That said, we agree with the reviewer that the accuracy of the reweighting approach will require further validation on a wider range of systems, which we believe is outside the scope of the work presented here. As the reviewer suggests, this could be tested by MD simulations of static proteins, but such a test would still assume that the protein-dye interactions are correctly captured in the force field, and hence could suffer from some of the issues discussed further above. Instead, we have opted to change the text to make it clearer that the accuracy and general utility of the reweighting approach needs further experimental (or computational) validation.

In the Conclusion section:

“In our case studies, this reweighting approach can result in better predictions compared to excluding frames with steric clashes, but its accuracy and general utility need further experimental and computational validation.”

I would like a brief discussion of how the rotamers are filtered and clustered to be included in the main text – it can be just a couple of sentences but this is important to the method. Also, the concepts of small medium and large libraries are not introduced before they are used in the results. This should be defined and ideally better supporting data for the choice of library size and how this influences accuracy should be provided or referenced.

We had initially removed detailed discussions from the main text to make the manuscript clearer, but agree that some text is useful. We now provide a short discussion of this in the main text referring also to the results presented in the supplement.

In the revised “Rotamer library Generation” subsection in the main text, we now outline the procedure used to generate the rotamer libraries:

“In FRETpredict, rotamer libraries are created through the following steps: (i) generation of the conformational ensemble of the FRET probe using MD simulations; (ii) selection of combinations of the most populated dihedral angles to generate the C1 set of cluster centers; (iii) assignment of trajectory frames to the

C1 set based on the least-square deviations of the dihedral angles; (iv) average over the angles of the trajectory frames of each C1 cluster center to generate the C2 set of cluster centers; (v) assignment of trajectory frames to the C2 set based on the least-square deviations of the dihedral angles; (vi) filtering of clusters with populations lower than 10, 20, and 30 structures to generate the rotamer libraries referred to as large, medium, and small hereafter. These steps are detailed in the Supporting Text and implemented in FRETpredict/rotamer_libraries.py.”

In the revised manuscript, the difference between large, medium, and small rotamer libraries is also mentioned at the beginning of the Results section:

“As detailed in Supporting Text 1, we generated rotamer libraries through a series of clustering steps, starting from large conformational ensembles from molecular simulations of the fluorophores in aqueous solutions. To further decrease the size of the rotamer libraries, we filtered out low-populated cluster centers based on three different cutoffs to obtain large, medium, and small rotamer libraries. To illustrate the extent to which the conformational ensemble of the probes is reduced upon the generation of the rotamer libraries, we plotted the projection on the xy-plane of the distance vectors between the Ca atom and the central atom of the fluorophore (Fig 2 and S4, S5, and S6) of all the generated rotamer libraries (Figs. S1, S2, and S3).”

Finally, in the caption of Fig 2, we give an estimate of the number of structures in the large rotamer libraries and refer to Fig. S7:

“2D projections of the position of the fluorophore with respect to the Ca atom for the large rotamer libraries generated in this work, which typically contain hundreds of structures (Fig. S7).”

We also mention the median number of rotamers in Fig. S7.:

“The median number of structures in the large, medium, and small rotamer libraries are 586, 189, and 100, respectively.”

The influence of the rotamer library size on the predictions of the method is now discussed in the Conclusions:

“Large rotamer libraries may lead to greater accuracy but are also more computationally expensive than smaller libraries. On the other hand, both medium and small rotamer libraries are a good compromise between calculation time and accuracy when long simulation trajectories are used. However, using a small number of rotamer clusters (i.e, small rotamer libraries) may compromise the prediction of FRET efficiency, especially in case of tight placement at the labeled site, in which many rotamers may be excluded from the calculation due to probe-protein steric clashes. Therefore, we recommend using large rotamer libraries when the computational cost is not a limiting factor and medium libraries for larger conformational ensembles.”

The different averaging regimes are very well described and implemented in the code – the authors should be congratulated on this. In the conclusion it is stated that they recommend the static

regime for single structures. The reason for this is not clear to me. If the dynamics regime is better when you have an ensemble of slowly moving protein structures, wont this also be the best for a single protein structure?

The reviewer is correct, and we indeed find that the dynamic regime is mildly more accurate than the static regime also for the less dynamic proteins. We have changed the text accordingly.

We rephrased the paragraph “MalE” of “Case study 3”:

“The RMSE values associated with the averaging regimes over all single-frame structures of HiSiaP, SBD2, and MalE are 0.097 (Static), 0.094 (Dynamic), 0.141 (Dynamic+), and 0.086 (Average). Based on these results, we observe that even in the case of single-frame structures, the best predictions correspond to the Dynamic regime.”

Conclusion:

“The Dynamic regime has been shown to provide better agreement with experimental data for both protein conformational ensembles and single protein structures. In the absence of information about the different timescales, we find that simply averaging the results from the three regimes often leads to good agreement with experiments.”

Minor points:

The introduction starts by describing single molecule FRET experiments and the use of rotamer library models for these. In fact, it is just as useful for ensemble FRET measurements so I would recommend noting this.

We agree and have updated the Introduction to also include ensemble measurements.

“Förster Resonance Energy Transfer (FRET) is a well-established technique to measure distances and dynamics between two fluorophores [1, 2]. Single-molecule FRET (smFRET) and ensemble FRET have been broadly used to study protein and nucleic acid conformational states and dynamics [3– 5], binding events [6, 7], and intramolecular transitions [8, 9]. The high spatial (nm) and temporal (ns) resolutions enable FRET experiments to uncover individual species in heterogeneous and dynamic biomolecular complexes, as well as transient intermediates [10–15].”

“Concomitantly, the molecular-level insights into protein structural dynamics provided by MD simulations are routinely employed to aid the interpretation of a multitude of experimental approaches, including FRET measurements [14, 20]”

The references to prior direct MD simulations of FRET dyes attached to biomolecules are somewhat selective to those by the authors. There are many extensive studies by the labs of Grubmuller, Corry and others that could be mentioned. Perhaps this need not be exhaustive in

the introduction but only citing those by the current authors is not ideal.

We have expanded the references to prior work in the introduction.

The current software outputs the average FRET efficiency as plotted in the nice bar graphs. For smFRET experiments it would also be useful to display the FRET efficiency distributions that can directly be compared to the raw experimental data.

The FRETpredict software already provides the per-frame FRET efficiency distributions (averaged over the rotamers) for the different averaging regimes, saved as text files. In the FRET.py script, in the trajectoryAnalysis function:

```
# Save <E>_static distribution

np.savetxt(self.output_prefix+'-Es-{:d}-{:d}.dat'.format(self.residues[0],self.residues[1]),esta_avg)

# Save <E>_dynamic1 distribution

np.savetxt(self.output_prefix+'-Ed1-{:d}-{:d}.dat'.format(self.residues[0],self.residues[1]),edyn1_avg)

# Save <E>_dynamic2 distribution

np.savetxt(self.output_prefix+'-Ed2-{:d}-{:d}.dat'.format(self.residues[0],self.residues[1]), edyn2_avg)
```

These values can then be used to calculate distributions that can then be compared to experimental distributions (when these can be generated in ways that do not reflect shot noise).

Reviewer 3

Montepietra and coworkers introduced a package called FRETpredict, which is able to predict FRET efficiencies from ensembles of protein conformations. As the paper shows that it can utilize a rotamer library approach to describe the FRET probes covalently bound to the protein. And they have tested on different types of systems: a relatively structured peptide and three folded proteins (HiSiaP, SBD2, and MalE). Overall, the paper was presented in a clear format with methodology and case studies in each section. The github and tutorials are also well documented.

We are happy that the reviewer found our paper to be clear, interesting and useful.

I feel the statistics and software will benefit a lot to this field of prediction and would recommend publication with minor concerns as follows:

1. Looks like all the efficiency plots do not have an error bar, does it mean the plots are obtained within 1-time trial?

Yes, the current plots indicate the average across the ensembles (or in some cases single structures). As discussed in the answer to reviewer 2, we now also make it possible to output timeseries. These can in turn be used to generate error estimates, though how this is done depends on how the ensembles were generated.

2. The folded proteins are among transporters which means more membrane protein and the protein size looks not that big. Can the author explain why to chose them as case study proteins, and are they the standard protein designed as test cases for FRET prediction? Will larger membrane protein work for FRETpredict? also, will the authors try some soluble and important proteins as kinase for the test cases too?

In our choice of test systems, we selected proteins to display different levels of dynamics ranging from disordered proteins to more flexible folded proteins or relatively static proteins. We also selected systems that had carefully measured FRET data and had been validated using independent measurements. The systems cover a range of system sizes (up to 370 residues) and both FRETpredict and FRET experiments are applicable to large systems and distances. This is now discussed in the revised manuscript.

In particular, we updated the Introduction:

“In this work we introduce FRETpredict, an easy-to-use Python module based on the RLA that enables FRET efficiency calculations from protein conformational ensembles. We describe a general methodology to generate rotamer libraries for FRET probes and provide access to rotamer libraries for many commonly used dyes and linkers. We present case studies for proteins displaying different dynamics ranging from disordered proteins to flexible and relatively static folded proteins (ACTR, Polyproline 11, HiSiaP, SBD2, and MalE). We selected systems for which FRET data has been carefully measured and validated using independent

methods. The systems cover a size up to 370 residues (for MaleE), showing that both FRETpredict and FRET experiments are applicable to large systems and distances.”

We also added a sentence in “Case study 3”:

“The reference FRET efficiency data of this case study was obtained from experiments by Peter et al. [50], wherein Alexa Fluor 555 - C2R and Alexa Fluor 647 - C2R dyes were employed as donor and acceptor, respectively”.

REVIEWERS' COMMENTS:

Reviewer #1 (Remarks to the Author):

I am fully satisfied with the response of the authors to my initial (minor) comments and support publication of the manuscript in its present form. Gunnar Jeschke

Reviewer #2 (Remarks to the Author):

The authors have carefully considered the reviewer comments and have done a good job revising the paper. I recommend it is published.